# ONLINE LEARNING IN VARYING FEATURE SPACES WITH INFORMATIVE VARIATION

## ABSTRACT

Most conventional online learning literature implicitly assumed a static feature space, while in practice the feature space may vary over time with the emerging of new features and vanishing of outdated features, which is named as online learning with Varying Feature Space (VFS). There have been increasing attention that initiated the exploration into this novel online learning paradigm. However, none of them was aware of the potentially informative information embodied as presence / absence (i.e., variation in this paper) for each feature, which indicates that the existence of some features of the VFS can be correlated with the class labels. Such information can be potentially beneficial to predictive performance if properly used for the learning purpose. To this end, we formally formulate this specific learning scenario, namely Online learning in Varying Feature space with Informative Variation (OVFIV), and present a learning framework to address this problem. The essence of the framework aim for answering the following two questions: how to learn a model to capture the association of the existence of features with the class labels and how to incorporate such information into the prediction process in order to gain performance improvement. Theoretical analyses and empirical studies based on 17 datasets from diverse fields verify the validity of our proposed method.

## 1 INTRODUCTION

Classical online learning problems assume that the feature space used to learn a predictor remains static over time. However, this assumption does not hold in the setting of Varying Feature Space (VFS), where new features would join the learning process and old features would vanish over time. This emerging research field has numerous real-world applications, such as the healthcare monitoring (Yao et al., 2018). To monitor patients' health conditions using streaming data from various health devices, the feature space varies when new devices providing additional features are introduced and unnecessary devices removed, resulting in adding and diminishing features. The varying feature property arises from the dynamic addition and removal of devices over time.

To surmount this constraint, recent studies have done wide explorations, where VFS scenarios are categorized into three primary forms: 1) the trapezoidal feature space (Gu et al., 2022; Zhang et al., 2016; Liu et al., 2022; Gu et al., 2023), where data exhibits simultaneous growth in instance and feature numbers; 2) the evolving feature space (Hou et al., 2017; 2021; Lian et al., 2022), which restricts variations to be regular, focusing on knowledge transfer from vanishing to emerging features during overlapping periods; and 3) the capricious feature space (He et al., 2020; 2021b; Beyazit et al., 2019), which represents the general case encompassing trapezoidal and evolving situations, permitting feature set of adjacent points to differ arbitrarily. This study focuses on the third case.

Although previous efforts, existing studies have not yet noticed the potential information contained in the feature varying pattern of VFS. Specifically, the presence and absence of features can be relevant to the class labels, which is referred to as *informative variation* in our study. Returning to the healthcare application, the feature variation, i.e., whether a health monitor is utilized, can be relevant to the current patient's health condition. For instance, the presence of features associated with devices that are only used in severe cases may imply poor health condition of the patient; whereas their absence may indicate an improved health condition. In this sense, there may be potential benefits in incorporating such presence / absence properties of VFS in the model training process.

This paper aims to explore this novel problem, for which we call *Online learning in Varying Feature spaces with Informative Variation* (OVFIV), with prediction models trained on both the informative feature variation and the feature values themselves. We aim to provide a framework to lift existing works with feature variation information. There are two primary challenges: First, after formulating the informative variation, how to learn a classifier that captures useful variation? Note that not all feature variation would provide informative signals when the features are present in the VFS Constantly (e.g. essential health monitors such as blood pressure and heart rate), meaning their absence / presence provide no information and are thus irrelevant to the class label. Naively taking all feature variations into account would induces computational complexity and inefficiency. Second, how to incorporate such informative variation in classification models to help with the improvement of predictive performance? Simply concatenating features and their variation representations may not be viable, as this would mix distinct data concepts, potentially obstructing the learning process.

Therefore, we propose to build two base classifiers based on the binary variation representation space and original feature space individually, and then aggregate their predictions to form the final prediction of the framework. Specifically, we first learn a *sparse* classifier on the variation space (a.k.a. variation stream in this paper), for which the model weights are penalized via a sparse regularization. Introducing sparsity enables the model to perform well regardless of the amount of information in the variation space. Together with the base learner built based on the original feature space, we introduce an ensemble approach and its two variants to provide final predictions. The fundamental concept entails combining the predictions by weighted average, and then adaptively recalibrating their relative importance based on respective cumulative losses after each prediction. The variants aim to conquer limitations of the basic ensemble from diverse perspectives, thus can improve the performance in different datasets.

Experimental results based 17 datasets from various fields validate the robustness and validity of our proposed approaches. The contributions of this work are summarized as follow:

1. This is the first effort to formulate the informativeness of the variation space and supplement prediction models with this additional information.

2. We employ a sparse online learner to learn from the variation space, alleviating potential negative effects from non-informative feature variations.

3. We employ an adaptive ensemble approach to produce the final prediction. Our results demonstrate the effectiveness of our learning framework, rarely having negative effort to predictive performance.

The remainder of this paper is organized as follows. Section 2 presents related work. Section 3 formulates the research problem of VFS, specifies its attribute and presents the methods dealing with the two recognized challenges. Experimental setup, results and further discussions are presented in Section 4. Section 5 conclude this paper. We have also supplied related theoretical analysis and additional experimental results in the appendix.

## 2 RELATED WORK

Our framework is most related to online learning tasks under varying feature space (VFS). Current literature classifies VFS into three primary categories, as described in previous section: 1) the trapezoidal feature space (Gu et al., 2022; Zhang et al., 2016; Liu et al., 2022), 2) the evolving feature space (Hou et al., 2017; 2021; Alagurajah et al., 2020), and 3) the capricious feature space (He et al., 2020; Beyazit et al., 2019; You et al., 2023; He et al., 2021b;a; Schreckenberger et al., 2020; 2023). Because the first two VFS types impose rigid assumptions about feature variation, making it unsuitable for our problem, in this article, we focus our comparison exclusively on algorithms designed for capricious feature space. The following is a brief overview of these algorithms: He et al. (2020) discusses the generative approach of reconstructing the universal feature space which contains all possible features thus an online learner can be directly trained. Beyazit et al. (2019) gives the solution based on the idea of projection confidence. Schreckenberger et al. (2023) learns an interpretable random feature forests to handle varying feature space. Other three related works further explored VFS scenarios with constraints: semi-supervised and class imbalance learning (You et al., 2023), mixed-type data (He et al., 2021a) and incomplete supervision (He et al., 2021b). Although

widely discussed and applicable to our problem, however, because the neglecting of the information contained in feature variation, the performance of prior works still have improvement potentiality.

Another relevant topic is informatively missing observations or informative missingness, common in clinical trials and related domains. For instance, in a congestive heart failure study, exercise testing scheduled at 12 weeks may lack results for patients who died during the trial (Lachin, 1999), leading to missingness that can be related to other features' values. Several other works, including those by Liu et al. (2006) and Shih (2002), have also thoroughly investigated this issue. The key relevance to our study lies in informative feature presence or absence. However, some distinctions exist between our work and current efforts: 1) We concentrate on classification, exploiting additional information to improve predictions, whereas they focus more on imputation task aimed at reconstructing datasets; 2) Our online learning setting requires observing data points only once, in contrast to informative missingness scenarios without this limitation, rendering proposed offline algorithms incompatible. Therefore, these works are not applicable to our settings.

In summary, while existing works have explored related areas, none fully encapsulates our proposed problem and associated algorithms are not directly applicable. Therefore, this framework represents a novel formulation and solution method.

## 3 THE PROPOSED APPROACH

In this section, we first formally present the learning problem, and then detail the two components of our proposed methods individually.

### 3.1 PROBLEM FORMULATION

We formulate the learning process with a varying feature space based on He et al. (2021a) and He et al. (2020). Let $\{(\mathbf{x}_t, y_t)|t = 1, 2, ..., T\}$ denote an input sequence, where $\mathbf{x}_t = [x_1, x_2, ..., x_{d_t}]^\top \in \mathbb{R}^{d_t}$ is a $d_t$-dimensional vector observed at the $t$-th round, accompanied by a label $y_t \in \{-1, +1\}$. In a varying feature space, $d_i = d_j$ does not necessarily occur for any $i \neq j$. We can then construct a universal feature space using the sequence $\mathbf{x}_t$ until round $t$. Let $\mathcal{U}_t = \bigcup_{i=1}^t \mathbb{R}^{d_i}$ signify the universal feature space encompassing all emerged features up to round $t$. For missing features at $t$, i.e. those in $\mathcal{U}_t$ but not $\mathbb{R}^{d_t}$, we denote as $\mathbf{x}_M \in \mathbb{R}^M$. Similarly, for remaining features, denote the observed as $\mathbf{x}_O \in \mathbb{R}^O = \mathbb{R}^{d_t}$. Notably, $\mathbb{R}^O \cap \mathbb{R}^M = \emptyset$, hence the universal space decomposes into the direct sum of observed and missing subspaces: $\mathcal{U}_t = \mathbb{R}^O \oplus \mathbb{R}^M$.

The informative variation can then be formulated as the followings. Given $\mathbb{R}^O$, $\mathbb{R}^M$ and $\mathcal{U}_t$, we can obtain the position encoding of the missing features at round $t$. Let $\{\mathbf{m}_t|t = 1, 2..., T\}$ signify the corresponding variation sequence, where $\mathbf{m_t} = [m_1, m_2, ..., m_{|\mathcal{U}_t|}]^\top \in \mathcal{U}_t$. We have:

$$m_i = \begin{cases} 1 & \text{if } x_i \in \mathbb{R}^M \\ 0 & \text{if } x_i \in \mathbb{R}^O \end{cases}, i = 1, ..., |\mathcal{U}_t| \tag{1}$$

Thus, the variation sequence encodes whether each feature is missing at round $t$ with a fixed size feature space. In our setting, the probability that a feature $x_i$ is missing (i.e., the variation $\mathbf{m}_t$) may correlate with the class label. Formally, we term a feature $m_i$ of $\mathbf{m}_t$ informative if $P(m_i) \neq P(m_i|y_t)$, and $\mathbf{m}_t$ informative if $P(\mathbf{m}_t) \neq P(\mathbf{m}_t|y_t)$. Here $y_t$ denotes the true label of $x_t$. Informative variations provide an additional route to learn $y_t$ via Bayes' rule: $P(y_t|\mathbf{m}_t) = \frac{P(\mathbf{m}_t|y_t)P(y_t)}{P(\mathbf{m}_t)}$.

Given the feature stream $\mathbf{x}_t$ and variation stream $\mathbf{m}_t$, our objective becomes learning $y_t$ from $\mathbf{x}_t$ and $\mathbf{m}_t$, for $t = 1, 2, ..., T$. Specifically, let $\{(\mathbf{x}_t, \mathbf{m}_t, y_t) \in \mathbb{R}^{d_t} \times \mathcal{U}_t \times \{-1, +1\}|t = 1, 2, ..., T\}$. At round $t$, the learner $\phi_t$ observes $\mathbf{x}_t, \mathbf{m}_t$ and then predicts. An immediate loss reflecting the discrepancy between prediction and true label is incurred, prompting learner update to $\phi_{t+1}$. Our goal is finding $\phi_1, ..., \phi_T$ that accurately predict the sequence via empirical risk minimization (ERM): $\min_{\phi_1,...,\phi_T} \frac{1}{T} \sum_{t=1}^T \ell(y_t, \phi_t(\mathbf{x}_t, \mathbf{m}_t))$, where $\ell(\cdot, \cdot)$ is a convex loss metric such as square or logistic loss.

## 3.2 LEARNING A PREDICTOR IN THE VARIATION SPACE

In this subsection, we discuss the usage of online models in learning from variation space.

The primary limitation of existing approaches which neglects potential information in $\mathbf{m}_t$ makes it necessary to integrate it into learning process. According to our formulation of $\mathbf{m}_t$, it is reasonable that an arbitrary online learner on that stream works. However, the problem is not such trivial considering that not all features' variation assist prediction, e.g., the essential monitors always appear so that the corresponding feature's variation provides no information. Ignorance of this potential risk makes the trained model suffered from noise. On account of the advantages of sparse model, we conjecture introducing sparsity could raise a solution by sparsifying the weights whose feature has uninformative variations, thus causing the learner to disregard those weights and thereby handle the problem effectively.

To introduce sparsity, a straightforward approach is regularization. Regularization methods are well-studied, with different techniques offering distinct properties (Tian & Zhang, 2022). Through preliminary experiments, we find $L_1$ regularization well-suited here. Formally, the learning objective becomes:

$$\forall t, \quad \mathbf{w}_t = \operatorname*{argmin}_{\mathbf{w} \in S} \sum_{i=1}^t \ell_i(\mathbf{w}) + \|\mathbf{w}\|_1 \tag{2}$$

where $S$ is the decision space.

A plausible way is using online gradient descent (OGD) which is essentially the same as stochastic gradient descent in batch learning and it is easy to adapt by just adding one regularization penalty term. However, this does not work as expected since it will essentially never produce weights that are exactly zero. Existing literature has explored online sparse learning problem extensively. In particular, Our target of learning an efficient and sparse learner can be tackled by online convex optimization (OCO). Existing works such as RDA (Xiao, 2009), FOBOS (Singer & Duchi, 2009), FTRL-Proximal (McMahan et al., 2013) all focus on the OCO problem. In this article, we use FTRL-Proximal as the classifier on variation feature space.

Denote a sequence of gradients as $\mathbf{g}_t \in \mathbb{R}^d$ and $\mathbf{g}_{1:t} = \sum_{s=1}^t \mathbf{g}_s$. The update rule of FTRL-Proximal is:

$$\mathbf{w}_{t+1} = \arg\min_{\mathbf{w}} \left( \mathbf{g}_{1:t} \cdot \mathbf{w} + \frac{1}{2} \sum_{s=1}^t \sigma_s \|\mathbf{w} - \mathbf{w}_s\|_2^2 + \lambda_1 \|\mathbf{w}\|_1 \right) \tag{3}$$

where $\sigma_s$ relates to the learning rate schedule by $\sigma_{1:t} = \frac{1}{\eta_t}$. With $\lambda_1 = 0$, this reduces to online gradient descent. Set the derivative of the term in $\arg\min$ to 0 and solve for $\mathbf{w}_{t+1}$ and it becomes $\mathbf{w}_{t+1} = \mathbf{w}_t - \eta_t \mathbf{g}_t$, where $\eta_t$ is a non-increasing learning-rate schedule, e.g., $\sigma_{1:t} = \frac{1}{\eta_t}$. For $\lambda_1 > 0$, sparsity induction performs well, as later experiments demonstrate.

Rewriting the $\arg\min$ term in Eq. (3), we obtain:

$$\left( \mathbf{g}_{1:t} - \sum_{s=1}^t \sigma_s \mathbf{w}_s \right) \cdot \mathbf{w} + \frac{1}{\eta_t} \|\mathbf{w}\|_2^2 + \lambda_1 \|\mathbf{w}\|_1 + (\text{const}) \tag{4}$$

Thus, storing $\mathbf{z}_{t-1} = \mathbf{g}_{1:t-1} - \sum_{s=1}^{t-1} \sigma_s \mathbf{w}_s$, at the beginning of round $t$ we update via $\mathbf{z}_t = \mathbf{z}_{t-1} + \mathbf{g}_t + (\frac{1}{\eta_t} - \frac{1}{\eta_{t-1}})\mathbf{w}_t$, and solve for $\mathbf{w}_{t+1}$ in closed form per-coordinate as:

$$w_{t+1,i} = \begin{cases} 0 & \text{if } |z_{t,i}| \le \lambda_1 \\ -\eta_t(z_{t,i} - \text{sgn}(z_{t,i})\lambda_1) & \text{otherwise.} \end{cases} \tag{5}$$

where $z_{t,i}$ denotes the $i$th component of $\mathbf{z}_t$, case of $\mathbf{w}_{t,i}$ is similar. The full procedure is summarized in Algorithm 1.

## 3.3 ENSEMBLE PREDICTION

In this subsection, we present one ensemble approach called OVFIV as well as two variants OVFIV-co and OVFIV-ca for aggregating predictions from the feature classifier and the variation classifier.

---

**Algorithm 1:** Learning from Feature Variation

---

**Data:** Input Sequence $\{(\mathbf{m}_t, y_t) | t = 1, 2, ..., T\}$
**Result:** Predictions $\{\mathbf{w}_t^\top \mathbf{m}_t | t = 1, 2, ..., T\}$

1 **for** $t = 1$ **to** $T$ **do**
2      Receive $\mathbf{m}_t$ and predict label $f_t = \mathbf{w}_t^\top \mathbf{m}_t$;
3      Suffer loss $\ell(f_t, y_t)$ and get gradient $\mathbf{g}_t \in \mathbb{R}^d$;
4      Update weights using Eq. (5) according to the gradient;
5 **end**

---

In our setting, where two synchronous streams are available, actually numerous approaches could potentially leverage the double data sources beyond learning separate predictors and ensembling. Since the streams align synchronously, the naive method is concatenating the feature spaces and yielding an even larger space. However, this approach merges heterogeneous streams with distinct perspectives, obstructing classifier optimization. Alternative approaches, such as online multi-view learning, particularly two-view methods (Nguyen et al., 2012), are also inapplicable, despite their ability to handle heterogeneous streams. Multi-view learners typically deal with data from multiple sources, such as a webpage's content text and link graph. Seemingly, our framework seems well-matched to such algorithms. However, fundamental distinctions arise: multi-view learning combines distinct views, whereas our variation stream offers supplementary information, rather than an additional view, to facilitate the overall learning process. The variation itself does not constitute data, hence cannot be regarded as a distinct view.

Instead, ensemble approaches seem the most viable strategy in our context, albeit not all online types apply here. Online bagging and boosting (Oza & Russell, 2001), for instance, are unsuitable as they rely on identical data sources, whereas our approach involves two classifiers operating on distinct streams. Furthermore, these ensemble methods employ numerous base learners collectively, whereas our approach utilizes only two. However, online learning with expert advice (Cesa-Bianchi & Lugosi, 2006) overcomes these limitations, and our methods build upon this technique.

Let the classifier predictions be:

$$f_{O,t} = \psi(\mathbf{x}_t), f_{M,t} = \mathbf{w}_{M,t}^\top \mathbf{m}_t \tag{6}$$

Since existing methods have given a solution for learning $\psi(\cdot)$, we use a general formulation and have no assumption about it. Our ensemble approaches generate final predictions per round using $f_{O,t}$ and $f_{M,t}$. Empirically and theoretically, the proposed methods can be shown matching the best individual component performance.

OVFIV combines predictions via weighted averaging:

$$\widehat{p}_t = \alpha_{O,t} f_{O,t} + \alpha_{M,t} f_{M,t} \tag{7}$$

With no preference, $\alpha$ initializes to $\frac{1}{2}$; otherwise, $\alpha$ reflects prior performance knowledge.

To minimize cumulative loss $L = \sum_{t=1}^T \ell(f_t, y_t)$, where $\ell(\cdot, \cdot)$ is the loss function, $f_t$ the prediction, and $y_t$ the label, updating $\alpha$ based on performance up to the current time step is reasonable:

$$\begin{cases} \alpha_{O,t+1} = \frac{e^{-\eta L_{O,t}}}{W_t} \\ \alpha_{M,t+1} = \frac{e^{-\eta L_{M,t}}}{W_t} \\ W_t = e^{-\eta L_{O,t}} + e^{-\eta L_{M,t}} \end{cases} \tag{8}$$

where $\eta$ is a tuned parameter and $W_t$ is the normalization term. $L_{p,t}$ denotes the cumulative loss of base learner $p \in \{O, M\}$ until round $t$. Intuitively, this updates $\alpha$ to weight learners exponentially based on relative suffered loss up to round $t$.

There are three approaches for setting hyperparameter $\eta$, the simplest ensemble OVFIV sets an empirically good single $\eta$ across datasets:

$$\eta = c, c > 0 \tag{9}$$

However, a fixed $\eta$ risks instability as it also depends on other factors. Fortunately, by Theorem 1, the optimal $\eta$ is:

$$\eta = \sqrt{\frac{8 \ln 2}{T}} \tag{10}$$

where $T$ is the stream size. This variant is termed OVFIV-c(ombination)o(ptimal). If $T$ is known a priori, e.g. for large offline data in a single-pass online setting, OVFIV-co surpasses empirical OVFIV theoretically. However, when the optimal $\eta$ is unavailable, a practical alternative is needed. To address this, the adaptive OVFIV-c(ombination)a(daptive) can be utilized, adjusting $\eta$ via:

$$\eta_t = \sqrt{8 \ln 2 / t} \tag{11}$$

Since $\eta_t$ depends only on $t$, this variant is practical. It also enjoys low regret bounds (Theorem 2). Algorithm 2 summarizes OVFIV and its two variants, OVFIV-co and OVFIV-ca.

---

**Algorithm 2:** OVFIV

**Data:** Input Sequence $\{(\mathbf{x}_t, \mathbf{m}_t, y_t) | t = 1, 2, ..., T\}$
**Result:** Predictions $\{\phi_t(\mathbf{x}_t, \mathbf{m}_t) | t = 1, 2, ..., T\}$

1 $\alpha_{O,1} = \alpha_{M,1} = \frac{1}{2}; L_{O,t} = L_{M,t} = 0;$
2 **for** $t = 1$ **to** $T$ **do**
3 $\quad$ Receive $\mathbf{x}_t \in \mathbb{R}^O, \mathbf{m}_t \in \mathbb{R}^M$ and predict $f_{O,t} = \psi(\mathbf{x}_t), f_{M,t} = \mathbf{w}_{M,t}^\top \mathbf{m}_t;$
4 $\quad$ Predict $\widehat{p}_t$ using Eq. (7), then receive the target $y_t$.
5 $\quad$ Suffer loss $\ell(\widehat{p}_t, y_t), \ell(f_{O,t}, y_t)$ and $\ell(f_{M,t}, y_t);$
6 $\quad$ Compute the cumulative loss $L_{O,t+1} = L_{O,t} + \ell(f_{O,t}, y_t), L_{M,t+1} = L_{M,t} + \ell(f_{M,t}, y_t)$
7 $\quad$ Update weights using Eq. (8), where $\eta$ is determined by Eq. (9), Eq. (10) or Eq. (11);
8 $\quad$ Update $\psi$ and update $\mathbf{w}_{M,t}$ by Algorithm 1;
9 **end**

---

## 4 EXPERIMENTAL STUDIES

In this section, we first introduce the experimental setup and comparative methods. Next, we present results obtained on various datasets. Finally, we study sparsity effects in our method.

### 4.1 EXPERIMENT SETTING

#### 4.1.1 DATASET

We assess the effectiveness of our algorithms using a diverse set of datasets, including 16 UCI datasets (Asuncion & Newman, 2007) spanning various domains and one real-world streaming dataset. To ensure controlled experiment and reduce interference, we balance highly imbalanced dataset through oversampling. Some UCI datasets are adapted from prior researches (You et al., 2023; He et al., 2021b), and are randomly shuffled with a fixed seed to simulate streaming data. The electricity dataset, derived from real streaming data, exposes our methods to more realistic conditions. Table 1 provides details regarding dataset sizes, the number of features, and imbalance ratios for all datasets.

Because all datasets are originally complete, to simulate varying feature spaces, we employ the following approach to generate semi-artificial datasets:

- For non-informative cases, all features of a data point exhibit a 50% missing rate, irrespective of the class label.

- In the case of informative scenarios, half of the features have a 50% missing rate, again regardless of the class label. The remaining features follow the conditions: $p(m_i | y_t = 0) = a$ and $p(m_i | y_t = 1) = b$, where missing rates are $a$ for one class and $b$ for the other class. Specifically, in our experimental study, we set $a = 0.1$ and $b = 0.3$.

Table 1: Characteristics of the studied datasets

| Dataset | #Inst | #Feat | IR | Dataset | #Inst | #Feat | IR |
|---|---|---|---|---|---|---|---|
| abalone | 4177 | 8 | 1.01 | glioma | 839 | 23 | 1.38 |
| adult | 5000 | 13 | 1.1 | wpbc | 302 | 34 | 1 |
| australian | 690 | 14 | 1.25 | ionosphere | 450 | 34 | 1 |
| credit-a | 653 | 15 | 1.21 | kr-vs-kp | 3196 | 36 | 1.09 |
| diabetes | 1000 | 8 | 1 | spambase | 4601 | 57 | 1.54 |
| diabetes-risk | 520 | 17 | 1.6 | splice | 3175 | 60 | 1.08 |
| dropout | 4424 | 36 | 1 | wbc | 916 | 9 | 1 |
| electricity | 5000 | 8 | 1.57 | wdbc | 569 | 30 | 1.68 |
| german | 1400 | 24 | 1 | | | | |

### 4.1.2 COMPETING METHODS

To ensure a fair evaluation, we employ two algorithms specifically designed for handling varying feature spaces, along with a naive model, as our feature classifiers for comparison. These base models serve as the learner $\psi$ in Eq. (6).

- **Naive**: Adapted from FTRL-ADP (Huynh et al., 2018) directly. It serves as a baseline model, providing essential insights into our methods. Upon encountering incomplete data, the model first replaces all missing values with zeros and subsequently trains a linear model on the complete data.

- **OVFM** (He et al., 2021a): Designed to handle varying feature space amid arbitrary changes, OVFM can also manage mixed-type data. The core concept involves learning a multivariate joint distribution known as the Gaussian Copula for mixed-type data and imputing missing values in incoming data using this distribution. Its classifier is trained on both the observed feature space, with zero-padding for missing values, and the latent feature space of Gaussian Copula.

- **OLIDS** (You et al., 2023): This algorithm was developed to address both incomplete and imbalanced data. Incomplete data may exhibit arbitrary missing values, falling within the realm of varying feature space. The main idea is identifying the most informative features by adhering to the empirical risk minimization principle.

We select these two related works for two primary reasons. Firstly, they are the only ones that have made their code publicly available. Secondly, conducting a comprehensive study using these methods can empirically confirm that our framework is independent of the base models and is irrelevant to incremental tasks on varying feature spaces.

We use ensemble method OVFIV-ca here, and results for other variants are presented in the appendix. We configure the $L_1$ regularization coefficient $\lambda$ at 1 to induce sparsity. Regarding the hyperparameters of the two compared methods, since our datasets partially overlap with theirs, we refrain from altering their fine-tuned settings. We employ the square error as the loss function $\ell$, which yields superior results in our preliminary experiments.

### 4.1.3 PERFORMANCE EVALUATION

We evaluate the models based on cumulative error rate (CER), defined as CER $= (1/t) \sum_{i \leq t} \mathbb{I}(y_i \neq \text{sign}(\widehat{y_i}))$, where $\mathbb{I}(\cdot)$ takes the value of 1 if the argument is true and 0 otherwise. Each configuration is executed 10 times with different seeds to obtain average performance (mean) $\pm$ standard deviation (std). To detect statistically significant differences among two or more methods across datasets, we employ the Friedman test. In cases where the null hypothesis is rejected, Nemenyi post-hoc test (Demšar, 2006) is conducted.

### 4.2 EXPERIMENTAL RESULTS

In this subsection, we conduct experiments addressing three specific questions and present the corresponding results.

Table 2: Comparison with existing methods involving information in the variation stream. Significant differences compared to OVFIV-ca are indicated in bold.

| | Naive | | | OVFM | | | OLIDS | | |
|---|---|---|---|---|---|---|---|---|---|
| | variation-classifier | feature-classifier | ensemble | variation-classifier | feature-classifier | ensemble | variation-classifier | feature-classifier | ensemble |
| abalone | **0.325 ± 0.012** | 0.263 ± 0.009 | 0.261 ± 0.010 | **0.325 ± 0.012** | 0.243 ± 0.006 | 0.234 ± 0.007 | **0.325 ± 0.012** | 0.263 ± 0.003 | 0.219 ± 0.009 |
| adult | 0.293 ± 0.008 | **0.386 ± 0.007** | 0.292 ± 0.008 | **0.293 ± 0.008** | **0.285 ± 0.007** | 0.241 ± 0.007 | 0.293 ± 0.008 | **0.313 ± 0.005** | 0.255 ± 0.006 |
| australian | **0.296 ± 0.017** | **0.274 ± 0.013** | 0.239 ± 0.016 | **0.296 ± 0.017** | 0.212 ± 0.012 | 0.197 ± 0.013 | **0.296 ± 0.017** | 0.232 ± 0.015 | 0.190 ± 0.018 |
| credit-a | **0.305 ± 0.011** | 0.273 ± 0.014 | 0.241 ± 0.019 | **0.305 ± 0.011** | 0.225 ± 0.012 | 0.203 ± 0.011 | **0.305 ± 0.011** | 0.232 ± 0.008 | 0.199 ± 0.011 |
| electricity | 0.310 ± 0.006 | **0.341 ± 0.006** | 0.298 ± 0.006 | **0.310 ± 0.006** | 0.278 ± 0.012 | 0.259 ± 0.011 | **0.310 ± 0.006** | **0.303 ± 0.004** | 0.247 ± 0.007 |
| kr-vs-kp | 0.155 ± 0.006 | **0.277 ± 0.006** | 0.152 ± 0.005 | 0.155 ± 0.006 | **0.234 ± 0.006** | 0.145 ± 0.005 | 0.155 ± 0.006 | **0.242 ± 0.005** | 0.147 ± 0.006 |
| spambase | 0.102 ± 0.004 | **0.167 ± 0.007** | 0.098 ± 0.004 | **0.102 ± 0.004** | 0.093 ± 0.004 | 0.069 ± 0.002 | 0.102 ± 0.004 | **0.125 ± 0.003** | 0.093 ± 0.004 |
| splice | 0.100 ± 0.005 | **0.342 ± 0.008** | 0.100 ± 0.005 | 0.100 ± 0.005 | **0.328 ± 0.004** | 0.100 ± 0.005 | 0.100 ± 0.005 | **0.306 ± 0.009** | 0.099 ± 0.004 |
| wdbc | **0.214 ± 0.011** | 0.062 ± 0.007 | 0.062 ± 0.008 | **0.214 ± 0.011** | 0.052 ± 0.006 | 0.049 ± 0.005 | **0.214 ± 0.011** | 0.061 ± 0.006 | 0.054 ± 0.004 |
| diabetes-risk | 0.274 ± 0.013 | **0.430 ± 0.020** | 0.279 ± 0.012 | 0.274 ± 0.013 | **0.371 ± 0.010** | 0.274 ± 0.012 | **0.274 ± 0.013** | 0.185 ± 0.010 | 0.153 ± 0.014 |
| glioma | 0.238 ± 0.009 | **0.254 ± 0.011** | 0.198 ± 0.011 | **0.238 ± 0.009** | 0.193 ± 0.005 | 0.163 ± 0.010 | **0.238 ± 0.009** | 0.177 ± 0.009 | 0.148 ± 0.008 |
| dropout | 0.157 ± 0.004 | **0.225 ± 0.007** | 0.150 ± 0.002 | 0.157 ± 0.004 | **0.189 ± 0.004** | 0.139 ± 0.003 | 0.157 ± 0.004 | **0.214 ± 0.005** | 0.148 ± 0.003 |
| diabetes | **0.341 ± 0.012** | 0.305 ± 0.013 | 0.242 ± 0.012 | **0.341 ± 0.012** | 0.296 ± 0.009 | 0.238 ± 0.013 | **0.341 ± 0.012** | 0.285 ± 0.010 | 0.234 ± 0.012 |
| german | 0.216 ± 0.006 | **0.334 ± 0.008** | 0.208 ± 0.006 | 0.216 ± 0.006 | **0.327 ± 0.009** | 0.204 ± 0.006 | 0.216 ± 0.006 | **0.315 ± 0.007** | 0.201 ± 0.006 |
| ionosphere | **0.220 ± 0.018** | 0.175 ± 0.013 | 0.146 ± 0.015 | **0.220 ± 0.018** | 0.160 ± 0.009 | 0.136 ± 0.014 | **0.220 ± 0.018** | 0.166 ± 0.008 | 0.126 ± 0.014 |
| wbc | **0.339 ± 0.017** | 0.055 ± 0.005 | 0.055 ± 0.005 | **0.339 ± 0.017** | 0.051 ± 0.005 | 0.050 ± 0.005 | **0.339 ± 0.017** | 0.088 ± 0.006 | 0.083 ± 0.005 |
| wpbc | 0.246 ± 0.022 | **0.395 ± 0.014** | 0.253 ± 0.022 | 0.246 ± 0.022 | **0.366 ± 0.018** | 0.245 ± 0.016 | 0.246 ± 0.022 | **0.377 ± 0.018** | 0.235 ± 0.018 |
| average rank | **2.265** | **2.529** | 1.206 | **2.588** | **2.353** | 1.059 | **2.588** | **2.412** | 1 |

Table 3: Comparison with existing methods involving no information in the variation stream. In most cases, our methods exhibit performance that does not significantly deviate from the original methods.

| | Naive | | | OVFM | | | OLIDS | | |
|---|---|---|---|---|---|---|---|---|---|
| | variation-classifier | feature-classifier | ensemble | variation-classifier | feature-classifier | ensemble | variation-classifier | feature-classifier | ensemble |
| abalone | 0.432 ± 0.005 | 0.354 ± 0.010 | 0.356 ± 0.011 | 0.432 ± 0.005 | 0.283 ± 0.008 | 0.282 ± 0.008 | 0.432 ± 0.005 | 0.265 ± 0.004 | 0.257 ± 0.004 |
| adult | 0.496 ± 0.005 | 0.343 ± 0.006 | 0.345 ± 0.006 | 0.496 ± 0.005 | 0.332 ± 0.007 | 0.336 ± 0.008 | 0.496 ± 0.005 | 0.323 ± 0.004 | 0.326 ± 0.004 |
| australian | 0.476 ± 0.022 | 0.229 ± 0.012 | 0.232 ± 0.013 | 0.476 ± 0.022 | 0.219 ± 0.011 | 0.219 ± 0.014 | 0.476 ± 0.022 | 0.228 ± 0.013 | 0.240 ± 0.018 |
| credit-a | 0.486 ± 0.015 | 0.235 ± 0.017 | 0.236 ± 0.015 | 0.486 ± 0.015 | 0.224 ± 0.015 | 0.225 ± 0.012 | 0.486 ± 0.015 | 0.226 ± 0.012 | 0.245 ± 0.012 |
| diabetes | 0.503 ± 0.017 | 0.341 ± 0.010 | 0.346 ± 0.010 | 0.503 ± 0.017 | 0.331 ± 0.010 | 0.334 ± 0.010 | 0.503 ± 0.017 | 0.327 ± 0.007 | 0.338 ± 0.009 |
| diabetes-risk | 0.422 ± 0.011 | **0.342 ± 0.015** | 0.378 ± 0.020 | 0.422 ± 0.011 | 0.318 ± 0.015 | 0.353 ± 0.023 | 0.422 ± 0.011 | 0.218 ± 0.008 | 0.235 ± 0.009 |
| dropout | 0.475 ± 0.007 | 0.268 ± 0.008 | 0.268 ± 0.008 | 0.475 ± 0.007 | 0.235 ± 0.005 | 0.235 ± 0.005 | 0.475 ± 0.007 | 0.237 ± 0.004 | 0.241 ± 0.005 |
| electricity | 0.415 ± 0.006 | 0.387 ± 0.007 | 0.382 ± 0.006 | 0.415 ± 0.006 | 0.303 ± 0.006 | 0.304 ± 0.007 | 0.415 ± 0.006 | 0.295 ± 0.006 | 0.306 ± 0.005 |
| german | 0.510 ± 0.010 | 0.378 ± 0.009 | 0.384 ± 0.008 | 0.510 ± 0.010 | 0.364 ± 0.007 | 0.371 ± 0.008 | 0.510 ± 0.010 | 0.353 ± 0.005 | 0.358 ± 0.008 |
| glioma | 0.475 ± 0.020 | 0.230 ± 0.009 | 0.233 ± 0.012 | 0.475 ± 0.020 | 0.217 ± 0.010 | 0.219 ± 0.010 | 0.475 ± 0.020 | 0.224 ± 0.009 | 0.240 ± 0.009 |
| ionosphere | 0.505 ± 0.017 | 0.232 ± 0.017 | 0.238 ± 0.017 | 0.505 ± 0.017 | 0.205 ± 0.015 | 0.208 ± 0.011 | 0.505 ± 0.017 | 0.212 ± 0.011 | 0.236 ± 0.013 |
| kr-vs-kp | 0.497 ± 0.005 | 0.282 ± 0.008 | 0.282 ± 0.007 | 0.497 ± 0.005 | 0.265 ± 0.007 | 0.266 ± 0.006 | 0.497 ± 0.005 | 0.259 ± 0.005 | 0.262 ± 0.006 |
| spambase | 0.458 ± 0.008 | 0.172 ± 0.003 | 0.172 ± 0.003 | 0.458 ± 0.008 | 0.164 ± 0.004 | 0.164 ± 0.003 | 0.458 ± 0.008 | 0.150 ± 0.003 | 0.153 ± 0.002 |
| splice | 0.501 ± 0.010 | 0.330 ± 0.008 | 0.331 ± 0.008 | 0.501 ± 0.010 | 0.325 ± 0.008 | 0.327 ± 0.009 | 0.501 ± 0.010 | 0.287 ± 0.006 | 0.291 ± 0.007 |
| wbc | 0.502 ± 0.019 | 0.075 ± 0.006 | 0.075 ± 0.006 | 0.502 ± 0.019 | 0.065 ± 0.007 | 0.065 ± 0.007 | 0.502 ± 0.019 | 0.106 ± 0.008 | 0.111 ± 0.009 |
| wdbc | 0.420 ± 0.015 | 0.066 ± 0.006 | 0.066 ± 0.007 | 0.420 ± 0.015 | 0.059 ± 0.010 | 0.060 ± 0.009 | 0.420 ± 0.015 | 0.074 ± 0.008 | 0.086 ± 0.008 |
| wpbc | 0.452 ± 0.020 | 0.421 ± 0.016 | 0.418 ± 0.021 | 0.452 ± 0.020 | 0.405 ± 0.024 | 0.419 ± 0.034 | 0.452 ± 0.020 | 0.416 ± 0.017 | 0.407 ± 0.023 |
| average rank | 3 | 1.265 | 1.735 | 3 | 1.176 | 1.824 | 3 | 1.118 | 1.882 |

**If the variation contains information, does our approach outperform existing works?**

According to Table 2, it is evident that when there is sufficient additional information to exploit, our approach consistently outperforms the individual base learners. The Friedman tests, conducted at a significance level of $0.05$, reject the null hypothesis (H0) with $p$-values of $0.00016$ (Naive), $6.94e-6$ (OVFM), and $2.54e-6$ (OLIDS), indicating significant differences between the methods. The average rank in last row provides a summary of relative performance. Notably, the ensemble achieves lowest rank, indicating its superiority among all competing methods. To further substantiate its superiority, the ensemble method is selected as the control for post-hoc tests, demonstrating its significant outperformance compared to the base classifiers. This reaffirms that the additional information indeed enhances overall performance.

**If the variation does not contain information, does our approach still maintain tolerable performance compared to existing algorithms?**

Table 3 presents the results of experiments in which the variation does not contain information, and the variation learner provides nearly nonsensical predictions in a balanced dataset setting. Notably, the enhanced version does not significantly lag behind the original classifiers. Specifically, Friedman tests conducted at a significance level of $0.05$ reject the null hypothesis (H0) with $p$-values of $3.82e-7$, $1.98e-7$ and $2.42e-7$ for the three cases with different feature classifiers respectively, signifying significant differences between methods. However, post-hoc tests reveal no significant difference between the feature classifier and the ensemble, with $p$-values of $0.36$, $1.98$, and $2.42$ for the three feature classifiers respectively. Consequently, our method maintains tolerable performance compared to the existing work.

**Does introducing sparsity crucial to our method?**

We conduct a comparison by varying the $L_1$ regularization coefficient settings to explore the importance of introducing sparsity, with Naive method as feature classifier. Based on results in Table 4, it is evident that sparsity enhances the performance of the base model in general by examining

Table 4: Comparison between various sparsity levels, with the best-performing one highlighted in bold text. The Nemenyi post-hoc test reveals that, at an informative level, our method with $\lambda = 1$ significantly outperforms the non-sparse counterpart.

| | non-informative | | | informative | | |
|---|---|---|---|---|---|---|
| | $\lambda=0$ | $\lambda=1$ | $\lambda=3$ | $\lambda=0$ | $\lambda=1$ | $\lambda=3$ |
| abalone | **0.356 ± 0.011** | 0.356 ± 0.011 | 0.356 ± 0.012 | 0.261 ± 0.011 | 0.261 ± 0.010 | **0.261 ± 0.010** |
| adult | **0.345 ± 0.006** | 0.345 ± 0.006 | 0.345 ± 0.006 | **0.290 ± 0.007** | 0.292 ± 0.008 | 0.292 ± 0.008 |
| australian | 0.233 ± 0.012 | 0.232 ± 0.013 | **0.231 ± 0.012** | 0.242 ± 0.015 | 0.239 ± 0.016 | **0.239 ± 0.016** |
| credit-a | 0.240 ± 0.016 | **0.236 ± 0.015** | 0.237 ± 0.018 | 0.242 ± 0.016 | **0.241 ± 0.019** | 0.246 ± 0.017 |
| electricity | 0.382 ± 0.006 | 0.382 ± 0.006 | **0.381 ± 0.005** | 0.299 ± 0.006 | 0.298 ± 0.006 | **0.293 ± 0.008** |
| kr-vs-kp | 0.282 ± 0.007 | **0.282 ± 0.007** | 0.282 ± 0.007 | 0.156 ± 0.005 | 0.152 ± 0.005 | **0.152 ± 0.006** |
| spambase | 0.172 ± 0.003 | **0.172 ± 0.003** | 0.172 ± 0.003 | 0.102 ± 0.005 | 0.098 ± 0.004 | **0.097 ± 0.003** |
| splice | **0.330 ± 0.009** | 0.331 ± 0.008 | 0.333 ± 0.008 | 0.107 ± 0.006 | **0.100 ± 0.005** | 0.101 ± 0.005 |
| wdbc | 0.069 ± 0.007 | 0.066 ± 0.007 | **0.065 ± 0.007** | **0.062 ± 0.007** | 0.062 ± 0.008 | 0.062 ± 0.007 |
| diabetes-risk | **0.369 ± 0.022** | 0.378 ± 0.020 | 0.375 ± 0.016 | **0.279 ± 0.017** | 0.279 ± 0.012 | 0.289 ± 0.015 |
| glioma | 0.234 ± 0.011 | 0.233 ± 0.012 | **0.233 ± 0.012** | 0.202 ± 0.014 | **0.198 ± 0.011** | 0.201 ± 0.008 |
| dropout | 0.268 ± 0.008 | 0.268 ± 0.008 | **0.267 ± 0.007** | 0.153 ± 0.003 | 0.150 ± 0.002 | **0.148 ± 0.003** |
| diabetes | **0.344 ± 0.012** | 0.346 ± 0.010 | 0.346 ± 0.011 | 0.243 ± 0.012 | **0.242 ± 0.012** | 0.243 ± 0.011 |
| german | **0.381 ± 0.007** | 0.384 ± 0.008 | 0.382 ± 0.008 | 0.212 ± 0.005 | **0.208 ± 0.006** | 0.211 ± 0.007 |
| ionosphere | 0.238 ± 0.018 | 0.238 ± 0.017 | **0.237 ± 0.019** | 0.148 ± 0.013 | **0.146 ± 0.015** | 0.157 ± 0.014 |
| wbc | 0.076 ± 0.006 | 0.075 ± 0.006 | **0.075 ± 0.006** | 0.056 ± 0.006 | **0.055 ± 0.005** | 0.055 ± 0.005 |
| wpbc | **0.418 ± 0.013** | 0.418 ± 0.021 | 0.429 ± 0.022 | **0.248 ± 0.014** | 0.253 ± 0.022 | 0.281 ± 0.028 |
| average rank | 2.118 | 2.088 | 1.794 | **2.441** | 1.559 | 2.000 |

the average rank. Further more, Friedman tests, conducted at the significance level of 0.05, reject H0 with a $p$-value of 0.017 when the variation is informative, indicating significant differences between the methods. Subsequently, post-hoc tests demonstrate that with a properly chosen $\lambda = 1$, the method with $\lambda = 0$ has a significant difference to its non-sparse counterpart. Therefore, our method can benefit from sparsity in general and even get significantly improved with a fined-tuned sparsity coefficient, supporting our speculation.

## 4.3 FURTHER DISCUSSION

Our methods significantly outperform existing works, as demonstrated through experiments. However, the added variation stream actually provides no new information beyond the incomplete stream, which implicitly have already encoded variation through missing features. Further explanation is still needed as to why introducing this seemingly redundant stream impacts performance. We propose two explanations:

1. Some current approaches rely on imputation techniques such as He et al. (2021a), He et al. (2020) and You et al. (2023), filling in missing features using imputed values. Although this simplifies learning on the universal space, it fails to retain the potential variation information.

2. The variation space offers a more abstract representation of sample points, independent of feature values. Isolating this concept and training an additional model benefits the overall process, as a single learner capturing both feature values and variation notions simultaneously appears inefficient.

## 5 CONCLUSIONS

This paper investigated online learning in a dynamic feature space characterized by informative variation. The key challenge involves harnessing the informative variation to improve predictive performance. Our approach involves representing feature variation as a binary stream, applying sparse learning for robust information extraction, and incorporating this information using ensemble methods. The proposed ensemble method, denoted as OVFIV, along with its two variants, balances two base classifiers based on cumulative loss to make final predictions. Theoretical analyses illustrate that the introduced supplementary space has the capability to augment the performance of pre-existing methodologies. We substantiate our contributions through extensive experiments, comparing with existing approaches and analyzing the role of sparsity in our method.

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

## A  APPENDIX

### A.1  THEORETICAL ANALYSIS

In this section, we assess the performance of our methods using the concept of *regret*. Concretely, we present two theorems based on Cesa-Bianchi & Lugosi (2006) that establish the loss bounds for our ensemble methods. Building upon this foundation, we study the asymptotic properties and demonstrate that they exhibit an *asymptotically no-regret* behavior when compared to the two base learners.

**Theorem 1.** *Assume that the loss function $\ell$ is convex in its first argument and that it takes values in $[0, 1]$. For all $T > 1$ and for all $y_t \in \mathcal{Y}$ with $t = 1, 2, ..., T$. The cumulative loss $L$ of OVFIV satisfies*

$$L - \min(L^O, L^M) \leq \frac{\ln 2}{\eta} + \frac{T\eta}{8} \tag{12}$$

*In particular, with $\eta = \sqrt{\frac{8\ln 2}{T}}$, we get the upper loss bound of OVFIV-co:*

$$L - \min(L^O, L^M) \leq \sqrt{\frac{T\ln 2}{2}} \tag{13}$$

This theorem indicates that the long-term average performance of the ensemble will, at a minimum, be comparable to that of its two components. This observation is substantiated by the fact that:

$$\frac{1}{T}\left(L - \min(L^O, L^M)\right) \leq \frac{\sqrt{\frac{T\ln 2}{2}}}{T} \longrightarrow 0$$

as $T \longrightarrow \infty$. Consequently, as the number of iterations $T$ increases, the performance of OVFIV-co asymptotically matches the performance of the better of the two classifiers, a phenomenon referred to as *asymptotic no-regret*.

**Theorem 2.** *Assume that the loss function $\ell$ is convex in its first argument and takes values in $[0, 1]$. For all $n \geq 1$ and for all $y_1, ..., y_n \in \mathcal{Y}$, the regret of OVFIV-ca with time-varying parameter $\eta_t = \sqrt{8\ln 2/t}$ satisfies*

$$L - \min(L^O, L^M) \leq 2\sqrt{\frac{T}{2}\ln 2} + \sqrt{\frac{\ln 2}{8}}. \tag{14}$$

This theorem guarantees that the bound for OVFIV-ca differs from the bound for OVFIV-co by only a constant asymptotically. Similar analysis reveals that it also exhibits asymptotic no-regret behavior.

### A.2  ADDITIONAL EXPERIMENTS

**Does ensemble learning enhance classification performance?**

While the theoretical guarantees for ensemble, empirical validation in our settings is lacking. To this end, we conduct experiments using the cumulative loss rate (CLR, or average cumulative loss) $(1/t)\sum_{i \leq t} \ell(y_i, \widehat{y}_i)$ as well aforementioned CER. Table 2 illustrates that the ensemble performance generally outperforms both base classifiers, as indicated by the bold text in most cases. Even in the worst-case scenarios, where one classifier exhibits significantly inferior performance compared to the other or even provides no useful predictions, as seen in Table 3, our methods can still approximate the performance of the best classifier, consistent with the theorem's support. Therefore, as assessing the relative performances between the two classifiers beforehand is challenging and requires knowledge of the informativeness of the variation stream, our ensemble methods alleviate this requirement, enabling their application across different datasets and situations. More detailed performance trends are presented in Figure 1 and Figure 2.

**Comparison between ensemble variants**

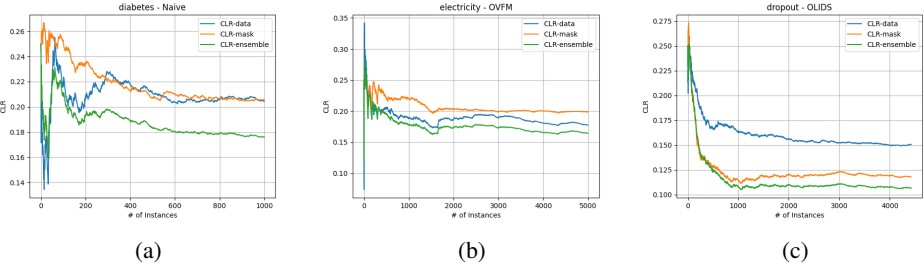

Figure 1: Ensemble performance CLR with different configuration

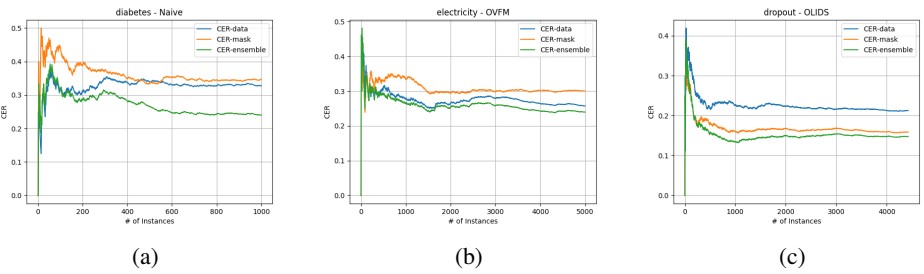

Figure 2: Ensemble performance CER with different configuration

In this subsection, we aim to assess the practical performance of ensemble method variants. We set the hyperparameters $\eta = 0.05$ for OVFIV. In this comparison, we keep other unrelated hyperparameters constant, specifically setting the $L_1$ regularization coefficient $\lambda$ to 1 and using the Naive method as the feature classifier. The results are presented in Table 5. From the results, we draw the following observations:

1. OVFIV's performance is not consistently stable across all datasets, consistent to our speculation. While it ranks first in many cases, its average performance does not exhibit distinct superiority.

2. OVFIV-co does not steadily outperform other methods, which seems contradictory to our theoretical analysis. We offer three explanations for this phenomenon: 1) The lower regret bound guarantees the worst performance but not actual performance; 2) The regret bound pertains to the loss function, which slightly differs from the cumulative error; and 3) The learning process is dynamic, with the performances of the two base learners continually changing, while the ensemble method strives to catch up to the best one. Consequently, a final guaranteed regret bound may not accurately reflect the performance on a limited-sized dataset. Instead, with a sufficiently large dataset, it provides more accurate predictions based on relatively stable experts, such as in the datasets electricity, kr-vs-kp and spambase.

3. OVFIV-ca generally performs better, thanks to its dynamic fine-tuning hyperparameter. Therefore, we recommend using OVFIV-ca with small sample datasets and OVFIV-co with sufficiently large sample datasets.

## A.3 Detailed proofs of theorems

In this section, we give a detailed proof about the strict regret bound of the ensemble methods by following the methods from Cesa-Bianchi & Lugosi (2006). For notation simplicity, we number our two learners as 1 and 2. That is we denote $L^O$, $L^M$ at round $t$ as $L_{1,t}$, $L_{2,t}$, $f_{M,t}$, $f_{O,t}$ as $f_{1,t}$, $f_{2,t}$, $\alpha_{O,t}$, $\alpha_{M,t}$ as $\alpha_{1,t}$, $\alpha_{2,t}$.

Table 5: Comparison between four ensemble methods. The best one is indicated by bold text.

| | non-informative | | | informative | | |
|---|---|---|---|---|---|---|
| | OVFIV-ca | OVFIV | OVFIV-co | OVFIV-ca | OVFIV | OVFIV-co |
| abalone | **0.356 ± 0.011** | 0.360 ± 0.014 | 0.359 ± 0.014 | **0.261 ± 0.010** | 0.264 ± 0.011 | 0.263 ± 0.011 |
| adult | 0.345 ± 0.006 | **0.345 ± 0.006** | 0.345 ± 0.006 | **0.292 ± 0.008** | 0.293 ± 0.008 | 0.293 ± 0.008 |
| australian | 0.232 ± 0.013 | 0.232 ± 0.013 | **0.231 ± 0.012** | 0.239 ± 0.016 | **0.239 ± 0.014** | 0.247 ± 0.013 |
| credit-a | **0.236 ± 0.015** | 0.238 ± 0.015 | **0.236 ± 0.015** | **0.241 ± 0.019** | 0.243 ± 0.016 | 0.247 ± 0.021 |
| diabetes | 0.346 ± 0.010 | 0.346 ± 0.010 | **0.344 ± 0.010** | 0.242 ± 0.012 | **0.241 ± 0.011** | 0.243 ± 0.012 |
| diabetes-risk | 0.378 ± 0.020 | **0.367 ± 0.019** | 0.390 ± 0.020 | **0.279 ± 0.012** | 0.286 ± 0.015 | 0.281 ± 0.013 |
| dropout | 0.268 ± 0.008 | 0.268 ± 0.008 | **0.267 ± 0.007** | **0.150 ± 0.002** | 0.152 ± 0.003 | 0.150 ± 0.003 |
| electricity | 0.382 ± 0.006 | 0.382 ± 0.006 | **0.381 ± 0.006** | 0.298 ± 0.006 | 0.299 ± 0.006 | **0.298 ± 0.006** |
| german | 0.384 ± 0.008 | 0.383 ± 0.009 | **0.383 ± 0.008** | **0.208 ± 0.006** | 0.210 ± 0.007 | 0.211 ± 0.006 |
| glioma | **0.233 ± 0.012** | 0.235 ± 0.013 | 0.234 ± 0.011 | **0.198 ± 0.011** | 0.200 ± 0.010 | 0.204 ± 0.010 |
| ionosphere | **0.238 ± 0.017** | 0.239 ± 0.019 | 0.239 ± 0.018 | 0.146 ± 0.015 | **0.140 ± 0.015** | 0.148 ± 0.014 |
| kr-vs-kp | 0.282 ± 0.007 | 0.282 ± 0.007 | **0.282 ± 0.007** | 0.152 ± 0.005 | 0.152 ± 0.005 | **0.152 ± 0.005** |
| spambase | **0.172 ± 0.003** | 0.172 ± 0.003 | 0.172 ± 0.003 | 0.098 ± 0.004 | 0.098 ± 0.004 | **0.097 ± 0.004** |
| splice | 0.331 ± 0.008 | **0.330 ± 0.008** | 0.330 ± 0.008 | **0.100 ± 0.005** | 0.101 ± 0.004 | 0.101 ± 0.005 |
| wbc | 0.075 ± 0.006 | **0.074 ± 0.006** | 0.075 ± 0.006 | 0.055 ± 0.005 | **0.054 ± 0.005** | 0.055 ± 0.005 |
| wdbc | **0.066 ± 0.007** | 0.069 ± 0.006 | 0.068 ± 0.007 | 0.062 ± 0.008 | 0.062 ± 0.007 | **0.062 ± 0.007** |
| wpbc | 0.418 ± 0.021 | **0.410 ± 0.022** | 0.430 ± 0.027 | **0.253 ± 0.022** | 0.257 ± 0.024 | 0.256 ± 0.021 |
| average rank | 2 | 2.147 | 1.853 | 1.5 | 2.176 | 2.324 |

### A.3.1 PROOF OF THEOREM 1

To prove Theorem 1, we begin by bounding the related quantities $(1/\eta) \ln(W_t/W_{t-1})$, recall

$$W_t = \sum_{i=1}^{2} \alpha_{i,t} = \sum_{i=1}^{2} e^{-\eta L_{i,t}}, \tag{15}$$

for $t \geq 1$, and $W_1 = 2$. $L_{i,t}$ is the cumulative loss at time $t$ of the $i$th base learner, namely $L_{i,t} = \sum_{t=1}^{T} \ell(f_{i,t}, y_t)$. Note that here $\alpha_{i,t}$ has not been normalized. In the proof we use the following classical inequality due to Hoeffding (Hoeffding, 1994).

**Lemma 1.** *Let $X$ be a random variable with $a \leq X \leq b$. Then for any $s \in \mathbb{R}$,*

$$\ln \mathbb{E}[e^{sX}] \leq s\mathbb{E}X + \frac{s^2(b-a)^2}{8}. \tag{16}$$

The detailed proof of Lemma 1 can be found in Section A.1 of the Appendix in Cesa-Bianchi & Lugosi (2006).

*Proof of Theorem 1.* First observe that

$$\begin{aligned}
\ln \frac{W_T}{W_1} &= \ln\left(\sum_{i=1}^{2} e^{-\eta L_{i,T}}\right) - \ln 2 \\
&\geq \ln\left(\max_{i=1,2} e^{-\eta L_{i,T}}\right) - \ln 2 \\
&= -\eta \min_{i=1,2} L_{i,T} - \ln 2.
\end{aligned} \tag{17}$$

On the other hand, for each $t = T_1 + 1, \ldots, T_1 + T_2$,

$$\begin{aligned}
\ln \frac{W_t}{W_{t-1}} &= \ln \frac{\sum_{i=1}^{2} e^{-\eta \ell(f_{i,t}, y_t)} e^{-\eta L_{i,t-1}}}{\sum_{j=1}^{2} e^{-\eta L_{j,t-1}}} \\
&= \ln \frac{\sum_{i=1}^{2} \alpha_{i,t-1} e^{-\eta \ell(f_{i,t}, y_t)}}{\sum_{j=1}^{2} \alpha_{j,t-1}}.
\end{aligned} \tag{18}$$

Now using Lemma 1, we observe that the quantity above may be upper bounded by

$$
-\eta \frac{\sum_{i=1}^{2} \alpha_{i,t-1} \ell(f_{i,t}, y_t)}{\sum_{j=1}^{2} \alpha_{j,t-1}} + \frac{\eta^2}{8}
$$
$$
\leq -\eta \ell \left( \frac{\sum_{i=1}^{2} \alpha_{i,t-1} f_{i,t}}{\sum_{j=1}^{2} \alpha_{j,t-1}}, y_t \right) + \frac{\eta^2}{8} \tag{19}
$$
$$
= -\eta \ell(\widehat{p}_t, y_t) + \frac{\eta^2}{8},
$$

where we used the convexity of the loss function in its first argument and the way how the weight updates. Summing over $t = 1, \ldots, T$, we get

$$
\ln \frac{W_T}{W_1} \leq -\eta L + \frac{\eta^2}{8} T \tag{20}
$$

Combining this with the lower bound and solving for $L$, we find that

$$
L \leq \min(L_{1,T}, L_{2,T}) + \frac{\ln 2}{\eta} + \frac{\eta}{8} T, \tag{21}
$$

as desired. In particular, with $\eta = \sqrt{8 \ln 2 / T}$, the upper bound becomes $\min(L_{1,T}, L_{2,T}) + \sqrt{(T/2) \ln 2}$. $\qquad \square$

### A.3.2 Proof of Theorem 2

**Lemma 2.** *For all $N \geq 2$, for all $\beta \geq \alpha \geq 0$, and for all $d_1, \ldots, d_N \geq 0$ such that $\sum_{i=1}^{N} e^{-\alpha d_i} \geq 1$,*

$$
\ln \frac{\sum_{i=1}^{N} e^{-\alpha d_i}}{\sum_{j=1}^{N} e^{-\beta d_j}} \leq \frac{\beta - \alpha}{\alpha} \ln N
$$

*Proof.* We begin by writing

$$
\ln \frac{\sum_{i=1}^{N} e^{-\alpha d_i}}{\sum_{j=1}^{N} e^{-\beta d_j}} = \ln \frac{\sum_{i=1}^{N} e^{-\alpha d_i}}{\sum_{j=1}^{N} e^{(\alpha-\beta) d_j} e^{-\alpha d_j}} = -\ln \mathbb{E} \left[ e^{(\alpha-\beta) D} \right] \leq (\beta - \alpha) \mathbb{E} D
$$

by Jensen's inequality, where $D$ is a random variable taking value $d_i$ with probability $e^{-\alpha d_i} / \sum_{j=1}^{N} e^{-\alpha d_j}$ for each $i = 1, \ldots, N$. Because $D$ takes at most $N$ distinct values, its entropy $H(D)$ is at most $\ln N$ (see Section A.2 of the Appendix in Cesa-Bianchi & Lugosi (2006)). Therefore,

$$
\ln N \geq H(D)
$$
$$
= \sum_{i=1}^{N} e^{-\alpha d_i} \left( \alpha d_i + \ln \sum_{k=1}^{N} e^{-\alpha d_k} \right) \frac{1}{\sum_{j=1}^{N} e^{-\alpha d_j}}
$$
$$
= \alpha \mathbb{E} D + \ln \sum_{k=1}^{N} e^{-\alpha d_k}
$$
$$
\geq \alpha \mathbb{E} D,
$$

where the last inequality holds because $\sum_{i=1}^{N} e^{-\alpha d_i} \geq 1$. Hence $\mathbb{E} D \leq (\ln N)/\alpha$. As $\beta > \alpha$ by hypothesis, we can substitute the upper bound on $\mathbb{E} D$ in the first derivation above and conclude the proof. $\qquad \square$

We are now ready to prove the Theorem 2.

*Proof of Theorem 2.* First, we study both $\ln(W_t/W_{t-1})$ and $\ln\left(w_{k_{t-1},t-1}/w_{k_t,t}\right)$, with the time-varying parameter $\eta_t$. Keeping track of the currently best expert, $k_t$ is the index of the expert with the smallest loss after the first $t$ rounds, and is used to lower bound the weight $\ln(w_{k_t,t}/W_t)$. In fact, the weight of the overall best expert (after $T$ rounds) could get arbitrarily small during the prediction process. We thus write the following:

$$\frac{1}{\eta_t}\ln\frac{w_{k_{t-1},t-1}}{W_{t-1}} - \frac{1}{\eta_{t+1}}\ln\frac{w_{k_t,t}}{W_t}$$
$$= \underbrace{\left(\frac{1}{\eta_{t+1}} - \frac{1}{\eta_t}\right)\ln\frac{W_t}{w_{k_t,t}}}_{(A)} + \underbrace{\frac{1}{\eta_t}\ln\frac{w'_{k_t,t}/W'_t}{w_{k_t,t}/W_t}}_{(B)} + \underbrace{\frac{1}{\eta_t}\ln\frac{w_{k_{t-1},t-1}/W_{t-1}}{w'_{k_t,t}/W'_t}}_{(C)}.$$

where $W'_t = \sum_{i=1}^{2} w_{i,t-1}e^{-\eta_t\ell(f_{i,t},y_t)}$ and $w'_{k_t,t} = e^{-\eta_t L_{k_t,t}}$. We now bound separately the three terms on the right-hand side. The term $(A)$ is easily bounded by noting that $\eta_{t+1} < \eta_t$ and using the fact that $k_t$ is the index of the expert with the smallest loss after the first $t$ rounds. Therefore, $w_{k_t,t}/W_t$ must be at least $1/2$. Thus we have

$$(A) = \left(\frac{1}{\eta_{t+1}} - \frac{1}{\eta_t}\right)\ln\frac{W_t}{w_{k_t,t}} \le \left(\frac{1}{\eta_{t+1}} - \frac{1}{\eta_t}\right)\ln 2.$$

We proceed to bounding the term $(B)$ as follows:

$$(B) = \frac{1}{\eta_t}\ln\frac{w'_{k_t,t}/W'_t}{w_{k_t,t}/W_t} = \frac{1}{\eta_t}\ln\frac{\sum_{i=1}^{2}e^{-\eta_{t+1}\left(L_{i,t}-L_{k_t,t}\right)}}{\sum_{j=1}^{2}e^{-\eta_t\left(L_{j,t}-L_{k_t,t}\right)}}$$
$$\le \frac{\eta_t - \eta_{t+1}}{\eta_t\eta_{t+1}}\ln 2 = \left(\frac{1}{\eta_{t+1}} - \frac{1}{\eta_t}\right)\ln 2,$$

where the inequality is proven by applying Lemma 2 with $d_i = L_{i,t} - L_{k_{t+1},t}$. Note that $d_i \ge 0$ because $k_t$ is the index of the expert with the smallest loss after the first $t$ rounds and $\sum_{i=1}^{2}e^{-\eta_{t+1}d_i} \ge 1$ as for $i = k_{t+1}$ we have $d_i = 0$. The term $(C)$ is first split as follows:

$$(C) = \frac{1}{\eta_t}\ln\frac{w_{k_{t-1},t-1}/W_{t-1}}{w'_{k_t,t}/W'_t} = \frac{1}{\eta_t}\ln\frac{w_{k_{t-1},t-1}}{w'_{k_t,t}} + \frac{1}{\eta_t}\ln\frac{W'_t}{W_{t-1}}.$$

We treat separately each one of the two subterms on the right-hand side. For the first one, we have

$$\frac{1}{\eta_t}\ln\frac{w_{k_{t-1},t-1}}{w'_{k_t,t}} = \frac{1}{\eta_t}\ln\frac{e^{-\eta_t L_{k_{t-1},t-1}}}{e^{-\eta_t L_{k_t,t}}} = L_{k_t,t} - L_{k_{t-1},t-1}.$$

For the second subterm, we proceed similarly to the proof of Theorem 1 by applying Hoeffding's bound (Lemma 1) to the random variable $Z$ that takes the value $\ell(f_{i,t},y_t)$ with probability $w_{i,t-1}/W_{t-1}$ for each $i = 1,2$:

$$\frac{1}{\eta_t}\ln\frac{W'_t}{W_{t-1}} = \frac{1}{\eta_t}\ln\sum_{i=1}^{2}\frac{w_{i,t-1}}{W_{t-1}}e^{-\eta_t\ell(f_{i,t},y_t)}$$
$$\le -\sum_{i=1}^{2}\frac{w_{i,t-1}}{W_{t-1}}\ell(f_{i,t},y_t) + \frac{\eta_t}{8}$$
$$\le -\ell(\widehat{p}_t,y_t) + \frac{\eta_t}{8}$$

where in the last step we used the convexity of the loss $\ell$. Finally, we substitute back in the main equation the bounds on the first two terms $(A)$ and $(B)$, and the bounds on the two subterms of the term $(C)$. Solving for $\ell(\widehat{p}_t,y_t)$, we obtain

$$\ell\left(\widehat{p}_t, y_t\right) \leq \left(L_{k_t,t} - L_{k_{t-1},t-1}\right) + \frac{\sqrt{a \ln 2}}{8} \frac{1}{\sqrt{t}}$$
$$+ \frac{1}{\eta_{t+1}} \ln \frac{w_{k_t,t}}{W_t} - \frac{1}{\eta_t} \ln \frac{w_{k_{t-1},t-1}}{W_{t-1}}$$
$$+ 2\left(\frac{1}{\eta_{t+1}} - \frac{1}{\eta_t}\right) \ln 2.$$

We apply the above inequality to each $t = 1, \ldots, T$ and sum up using $\sum_{t=1}^{T} \ell\left(\widehat{p}_t, y_t\right) = \widehat{L}_T$, $\sum_{t=1}^{T}\left(L_{k_t,t} - L_{k_{t-1},t-1}\right) = \min_{i=1,2} L_{i,T}$, $\sum_{t=1}^{T} 1/\sqrt{t} \leq 2\sqrt{T}$, and

$$\sum_{t=1}^{T}\left(\frac{1}{\eta_{t+1}} \ln \frac{w_{k_t,t}}{W_t} - \frac{1}{\eta_t} \ln \frac{w_{k_{t-1},t-1}}{W_{t-1}}\right) \leq -\frac{1}{\eta_1} \ln \frac{w_{k_1,1}}{W_1} = \sqrt{\frac{\ln 2}{a}}$$

$$\sum_{t=1}^{T}\left(\frac{1}{\eta_{t+1}} - \frac{1}{\eta_t}\right) = \sqrt{\frac{T+1}{a(\ln 2)}} - \sqrt{\frac{1}{a(\ln 2)}}.$$

Thus, we can write the bound

$$\widehat{L}_T \leq \min_{i=1,2} L_{i,T} + \frac{\sqrt{aT \ln 2}}{4} + 2\sqrt{\frac{(T+1) \ln 2}{a}} - \sqrt{\frac{\ln 2}{a}}.$$

Finally, by overapproximating and choosing $a = 8$ to trade off the two main terms, we get

$$\widehat{L}_T \leq \min_{i=1,2} L_{i,T} + 2\sqrt{\frac{T}{2} \ln 2} + \sqrt{\frac{\ln 2}{8}}$$

as desired. $\qquad\square$

