# OpenReview forum: "Online Learning in Varying Feature Spaces with Informative Variation"
_ICLR.cc/2024/Conference — ICLR 2024 Conference Withdrawn Submission_

### Official Review · Reviewer_FQaF · 2023-10-30

**Soundness:** 2 fair
**Presentation:** 2 fair
**Contribution:** 2 fair
**Rating:** 3
**Confidence:** 3

**Summary:**

This work studies an online setting where the received features might be presented with absent entries in different iterations. The authors propose a FTRL-based algorithm with 1-norm as the regularizer to learn the sparsity representation. Further they utilize an aggregating algorithm to combine the sparsity representation and the prediction (provided in a black-box way) for observed data. This work carries out experiments to validate their findings.

**Strengths:**

This work carries out detailed experiments to validate their findings and analyzes the results.

**Weaknesses:**

The writing and notation seem slightly unclear to me, and I find certain symbols introduced but not subsequently utilized within the text. The method of online learning proposed for $\mathbf{w}_t$ fails to yield novel insights as claimed in the intro. The approach of employing an aggregating algorithm to combine the decisions is common and the theoretical result is a direct extension of [Cesa-Bianchi et al., 2006].

**Questions:**

[Freund et al., 1997] proposed a framework, wherein certain experts might be absent in specific iterations. I think it is beneficial to compare with this work.

The theoretical result is obtained under the adversary assumptions, and the algorithm cannot adapt to the benefit of $L^O$ or $L^M$, as indicated by $O(\sqrt{T})$ gap in the regret bound.

The relationship $\Phi(x) = x$ seems to adopt a linear assumption for the online learner. The rationale behind employing Bayes' rule to examine sparsity is not immediately clear to me.

In equation (7), what is the justification for directly summing the representations from O space and M space?

Yoav Freund, Robert E Schapire, Yoram Singer, and Manfred K Warmuth. Using and combining predictors that specialize.

---

### Official Review · Reviewer_QVUf · 2023-10-31

**Soundness:** 2 fair
**Presentation:** 1 poor
**Contribution:** 3 good
**Rating:** 3
**Confidence:** 4

**Summary:**

The manuscript delves into the challenge of online learning within the context of Varying Feature Space (VFS), where the authors suppose that the existence of the features is related to the data labels. The authors posit that this relationship could be instrumental in enhancing model performance. In response to this challenge, a novel method amalgamating sparsity and ensemble learning is introduced to handle the VFS issue. A series of comprehensive experiments are conducted to substantiate the efficacy of the proposed methodology.

**Strengths:**

The paper delves into an intriguing setting, highlighting the understudied correlation between feature existence and data labels. The authors' insightful observation and approach to this correlation are commendable and contribute to the novelty of the work.

**Weaknesses:**

While the paper provides a keen insight into the VFS problem, it exhibits several shortcomings, particularly in the methodological, theoretical, and experimental segments. These issues are further delineated in the Questions section below.

Additionally, there is substantial room for enhancement in terms of the manuscript’s writing quality.

**Questions:**

1. The method delineated in Section 3.2 lacks clarity, particularly in how the sparsity can contribute to the filtering of uninformative feature variations. The authors are encouraged to furnish additional explanations and intuitive insights regarding their proposed method.
2. I am inclined to believe that the sparsity aspect might actually overlook the correlation between feature existence and data labels, given that it tends to favor features that are prevalently utilized by the classifier.
3. Theorem 1 appears to be somewhat meaningless, providing a regret guarantee solely on the empirical loss $L$, rather than on the expected loss. This could potentially lead to overfitting, failing to assure the classifier's performance.
4. For the experiments, the authors only conduct on simulated data. Can the authors find some real-world applications of the VFS problem?
5. The manuscript’s writing requires enhancements:

    * The use of symbols in the problem formulation is ambiguous; $\mathbb{R}^M$ typically denotes Euclidean space of $M$ dimension, not a specific distribution. It would be more appropriate to use $\mathcal{D}_M$  in this context. The same correction applies to $\mathbb{R}^O$.
    * Section 3, which explicates the proposed method, is prolix and lacks essential intuition. A revision for clarity and conciseness is needed.
    * Equation (10) and the approach that $\eta = O(1/\sqrt{T})$ can be omitted, as setting $\eta = O(1/\sqrt{t})$ is a conventional practice in the field of online learning.

---

### Official Review · Reviewer_xjMG · 2023-11-04

**Soundness:** 2 fair
**Presentation:** 1 poor
**Contribution:** 2 fair
**Rating:** 3
**Confidence:** 3

**Summary:**

The research introduces a novel framework for addressing the challenges posed by varying feature spaces in online learning environments. Recognizing that features in such spaces may come and go over time, the authors present a methodology called Online Learning in Varying Feature Spaces with Informative Variation (or OVFV). The OVFV framework is designed to adaptively exploit informative variation in feature spaces, which is particularly relevant in fields like healthcare monitoring where the feature set can change dynamically. The authors argue that their approach can improve learning performance by appropriately weighting the presence or absence of information, and provide empirical evidence of the effectiveness of their method across 17 datasets from diverse fields.

**Strengths:**

- The setting of handling varying feature spaces in online learning is not new, but the authors' approach to leveraging informative variation is novel to me. The concept of utilizing both the presence and absence of features as informative signals is creative and could open new directions for research in the field.

- The paper includes experimental results from 17 datasets, providing a substantial empirical basis for their claims.

**Weaknesses:**

1. The experimental section seems lack of some baselines. Feature-wise online learning is a field that has been extensively studied. The author should consider some classic baselines, such as a series of works on online feature selection. And most of the datasets in Table 1 have very small feature spaces. If the author could provide a simulation on a dataset with a significantly larger number of features than the sample size, it would be more convincing, especially in the context of this work's focus on online features.

2. The theoretical analysis in this manuscript is pretty elementary, and it would be great if the author could add some complexity comparison and convergence boundary analysis with current SOTA feature-wise online learning algorithms. Besides, there is no discussion on the scalability and theoretical limits of the proposed method.

3. The quality of the presentation is far from satisfaction. There are lots of the grammatical issues / typos in the current form, e.g,

- (page 1) "... and unnecessary devices removed" -> missing "are" before "removed"
- (page 2) "... feature variations into account would induces" -> should be "induce"
- (page 2) "Experimental results based 17 datasets..." -> add "on"
- (page 2) "... are summarized as follow:" -> should be " as follows"
- (page 2) "... having negative effort to" -> "negative effort" or "negative effect"?
- (page 2) "Section 5 conclude this paper" -> should be "concludes"
- (page 3) "... performance of prior works still have" -> should be "has"
- (page 4) "... which neglects potential information" -> should be "neglect"
- (page 4) "... performance of prior works still have" -> should be "has"
- (page 5) "Seemingly, our framework seems well ..." -> word "Seemingly" and "seems" are redundant
- ... (I stopped marking them after a while)

besides, there're also many long sentences appear awkward and don't convey a clear meaning. In the context of a scientific or technical paper, the precision and clarity are very important. Hence this manuscript needs significant revision before being accepted by any venues.

**Questions:**

A natural idea is to encode the existence and disappearance of features into a variable of $\\{0,1\\}^d$ ($d$ is the dimension), and then transform it into a known hidden variable model for solution or optimization. I am curious why the author did not draw inspiration from a series of works in statistics and optimization.